# Genomic Insights of First *erm*B-Positive ST338-SCC*mec*V_T_/CC59 Taiwan Clone of Community-Associated Methicillin-Resistant *Staphylococcus aureu**s* in Poland

**DOI:** 10.3390/ijms23158755

**Published:** 2022-08-06

**Authors:** Ksenia Szymanek-Majchrzak, Grażyna Młynarczyk

**Affiliations:** Department of Medical Microbiology, Medical University of Warsaw, Chalubinskiego 5 Str., 02-004 Warsaw, Poland

**Keywords:** CA-MRSA, *erm*B gene, IS1216V insertion sequence, macrolides lincosamides and streptogramin B (MLS-B), mobile element structure (MES PM1), Panton–Valentine leukocidin (PVL), ST338/CC59, Taiwan clone

## Abstract

We report the first Polish representative of community-associated methicillin-resistant *Staphylococcus aureus* (CA-MRSA), *luk*S/F-PV-positive, encoding the *erm*B gene, as a genetic determinant of constitutive resistance to macrolides, lincosamides, and streptogramin B antibiotics, cMLS-B. This is the first detection of the CA-MRSA strain responsible for nosocomial infection in the Warsaw Clinical Hospital. Resistance to β-lactams associates with a composite genetic element, SCC*mec* cassette type V_T_ (5C2&5). We assigned the strain to sequence type ST338 (single-locus variant of ST59), clonal complex CC59, *spa*-type t437, and *agr*-type I. Genomic-based comparison was designated SO574/12 as an international Taiwan clone, which has been so far described mainly in the Asia-Pacific region. The *erm*B gene locates on the chromosome within the 14,690 bp mobile element structure, i.e., the MES_PM1-like_ structure, which also encodes aminoglycoside- and streptothricin-resistance genes. The MES_PM1-like_ structure is a composite transposon containing Tn551, flanked by direct repeats of IS1216V insertion sequences, which probably originates from *Enterococcus*. The *erm*B is preceded by the 273 bp regulatory region that contains the regulatory 84 bp *erm*BL ORF, encoding the 27 amino acid leader peptides. The latest research suggests that a new leader peptide, *erm*BL2, also exists in the *erm*B regulatory region. Therefore, the detailed function of *erm*BL2 requires further investigations.

## 1. Introduction

Methicillin-resistant *Staphylococcus aureus* (MRSA) is described as one of the most dangerous human pathogens. For many years, community-associated MRSA, CA-MRSA, has been limited to populations outside healthcare settings. It was just in the last two decades that it were considered clinically significant and a potentially highly virulent pathogen associated with serious, highly invasive and progressive skin and soft tissue infections, necrotizing pneumonia, sepsis, and fasciitis, particularly in young healthy individuals [1,2,3]. In recent years, due to the evolving epidemiology of CA-MRSA, these strains have also emerged as the cause of hospital outbreaks. Nosocomial outbreaks of CA-MRSA have been reported in various parts of the world, often affecting particular areas of hospitals, such as neonatology, pediatric, obstetric, or ophthalmic units, where the prevalence of healthcare-acquired MRSA, HA-MRSA, has been low [1,3,4,5,6].

CA-MRSA strains differ from HA-MRSA. They have unique epidemiology, phylogenetic origin, and genetic profile that is linked with carrying a smaller version of SCC*mec* (mainly type IV and V or V_T_); an ability to produce toxins, e.g., Panton–Valentine leukocidin (PVL), which confer higher toxigenic potential; and expression of a unique antibiotic resistance pattern (resistance to fewer non-β-lactam antibiotics than HA-MRSA) [2,7].

In 2005, a new variant of SCC*mec* type V was identified in *S. aureus* strain TSGH17, belonging to ST59/CC59 [8]. This variant of chromosome cassette, which is a composite genetic element, has been called SCC*mec* type V_T_ [8,9,10]. Nowadays, it is also called SCC*mec* type Vb. Panton–Valentine leukocidin (PVL) is a bicomponent, β-barrel pore-forming leukotoxin [11]. Genes *luk*S-PV and *luk*F-PV encoding the subunits of the Panton–Valentine leukocidin are located on prophage ΦSA2, carried by 2–3% of *S. aureus* isolates. It has been proven that there is a correlation between the occurrence of *luk*S/F-PV genes and the type of SCC*mec* cassette [2,12,13,14].

CA-MRSA strains are usually limited to β-lactam resistance [15]. Recently, a higher number of CA-MRSA strains exhibiting a multidrug-resistant phenotype has been reported. Among them, isolates expressing the phenotype of resistance to macrolides, lincosamides, and streptogramin B (MLS-B) are common [16]. In CA-MRSA ST59/CC59 strains, the *erm*B gene has been found [17,18].

Resistance to macrolides, lincosamides, and streptogramin B (MLS-B) in *S. aureus* is expressed mainly as a result of a cross-resistance mechanism. The mechanism is related to the ability of resistant strains to produce Erm N6-dimethyltransferases, which, through adenine dimethylation (A2058) in the V 23S rRNA domain of the 50S ribosome subunit, make this site inaccessible to all MLS-B antibiotics [19]. In staphylococci*, erm*A and *erm*C genes are the most common and, until recently, have been described mainly in HA-MRSA strains. Whereas, *erm*B is much less frequent. Until now, the *erm*B gene has been described primarily in Gram-negative rods and has dominated in resistant variants of *Streptococcus* and *Enterococcus* [20,21,22,23].

In recent years, *erm*B MRSA strains arising from human clinical samples have been more frequently described in the worldwide literature. Surprisingly, the strains encoding the *erm*B gene represent CA-MRSA variants [10,22,23,24]. According to previous investigations, all MLS-B resistant isolates detected in the Warsaw Clinical Hospital represented Brazilian/Hungarian, Hannover, and Iberian clones, all of which had the HA-MRSA phenotype [21], between 1991–2007, and EMRSA-16 and USA200 clones (ST36/CC30), which also represented the HA-MRSA variants, between 2010–2011 [25]. All of these strains encoded *erm*A and/or *erm*C genes as determinants of MLS-B antibiotic resistance. During the period of 1991 to 2011, there was no CA-MRSA *erm*B-carrying strain among the MRSA isolates collected from surgical and transplant patients in the Warsaw Clinical Center [21,25]. Different studies conducted in Poland have not also reported any *erm*B-positive MRSA case, in recent years [1,26,27]. Surprisingly, in the preliminary analysis, an *erm*B-positive MRSA isolate was detected in one of clinical samples collected in 2012. This isolate has been recognized as a unique microbiological material on the Polish scale and is even rare in Europe. Therefore, we undertook a research project to characterize both the phenotype and the genotype, based on whole-genome sequencing—next generation sequencing (WGS-NGS) technology for a detailed analysis of genomic features. The main aims of our study were to describe the genetic structure and organization of the *erm*B-carrying genetic region within the surrounding genetic environment as well as characterize the virulome and resistome of the analyzed strain.

## 2. Results

### 2.1. Preliminary Phenotype and Genotype-Based Results

Preliminary phenotype and genotype analysis showed that the introduced strain SO574/12 was low-level methicillin-resistant *S. aureus* (MRSA) with the *mec*A gene, which determined resistance to almost all β-lactams. The *mec*A gene was carried within the SCC*mec* cassette type (subtype) Vb (5C2&5), cassette chromosome recombinase *ccr* genes complex type *ccr*C1-allele-2, and *ccr*C1-allele-8, *mec* gene complex class C2. The SO574/12 bacterial isolate was assigned to *spa*-type t437, *agr*-type I, and sequence type ST338, clonal complex CC59, and, according to SCC*mec* and MLST typing methods, was classified as an epidemic “Taiwan” clone. The strain expressed the phenotype of resistance to macrolides, lincosamides, and streptogramin B antibiotics (MLS-B) via constitutive mechanism of resistance (cMLS-B) in D-test zone method (all disc zones equal 6 mm, MIC value >256 mg/L). When using end-point PCR to detect *erm*A/B/C genes, only *erm*B gene was positively confirmed (amplicon was 359 bp in size). The main features of SO574/12 clinical isolate and details of antimicrobial resistance profile to 21 antimicrobials are listed in Table 1 and Table 2, parts A and B.

### 2.2. Genomic-Based Results

#### 2.2.1. Genome-Assembly Features

The MRSA *erm*B SO574/12 was assembled with the use of SPAdes. The coarse consistency of the genome was 100%, fine consistency was 99.8%, and completeness was 100%. There were 67 contigs (the largest 494,917 bp), an estimated genome length of 2,792,694 bp, and an average G+C content of 32.74%. The N50 length was 127,953 bp. The L50 count was 7. Key assembling data for MRSA SO574/12 isolate are presented in Table 3.

The DNA sequence of the above-mentioned CA-MRSA strain SO574/12 CC59/ST338/SCC*mec*Vb chromosome was deposited in the NCBI repository, GenBank accession number: JAJNOI000000000 (WGS submission: SUB10571769; BioProject: PRJNA774368; BioSample: SAMN22563086).

The CA-MRSA ZY05 SCC*mec*Vb/ST338/CC59 strain was used as a reference genome for comparative analysis. The general genome and protein features of the tested MRSA SO574/12 and reference CA-MRSA ZY05 SCC*mec*Vb/ST338/CC59 (GenBank accession number: CP045472.1) strains are presented in Table 4.

#### 2.2.2. Genome Annotation and Taxonomy Confirmation

The genome of the SO574/12 strain was annotated using the RAST tool kit (RASTtk) and assigned a unique genome identifier: 1280.32586.

This genome has 2643 protein coding sequences (CDS) and 60 transfer RNA (tRNA) genes (see Appendix A). The annotation included 512 hypothetical proteins and 2131 proteins with functional assignments. The proteins with functional assignments included 740 proteins with Enzyme Commission (EC) numbers, 613 with Gene Ontology (GO) assignments, and 534 proteins that were mapped to Kyoto Encyclopedia of Genes and Genomes (KEGG) pathways.

A circular graphical display of the distribution of the genome annotations is provided (Figure 1). Each color informs about a different CDS type. It includes rings, from outer to inner rings: the contigs, CDS on the forward strand, CDS on the reverse strand, RNA genes, CDS with homology to known antimicrobial resistance genes (AMR), CDS with homology to know virulence factors (VF), GC content, GC skew, and others.

#### 2.2.3. Comparative Genomics of CA-MRSA SO574/12

The global whole-genome and local small-scale alignment with the use of the progressive Mauve algorithm was performed for two genomes: CA-MRSA ZY05 Taiwan clone, as reference, and CA-MRSA SO574/12. The resulting alignment was visualized using the PATRIC BBRC website, providing insight into homologous regions and changes due to DNA recombination. The details of the global large-scale genome alignment are presented in Figure 2.

Each type of locally collinear block (LCB) was marked with a different color. LCBs containing highly conserved homology regions were assigned with the same stain. Homology regions that were defined within both aligned genomes were connected with a line in the same color. Thirty-six common LCBs were detected. The maximum and minimum lengths of the LCBs were 372,796 bp and 641 bp, respectively, in the ZY05 genome, and 372,878 bp and 643 bp, respectively, in the SO574/12 genome. The summary length of the LCBs was 2,761,725 bp in the ZY05 genome and 2,750,602 bp in the SO574/12 genome, which were 97.85% and 98.50% of the total genome sizes, respectively. The *erm*B gene was detected in contig 33 (size 14,772 bp) of the CA-MRSA SO574/12 genome. The size of the homology region in the genome of CA-MRSA ZY05 was 14,767 bp.

The local small-scale sequence alignment allows to compare homologous regions insight of defined pair of locally collinear blocks (LCBs), which encode the *erm*B gene for both CA-MRSA strains. Additionally, plasmid sequence pEflis48 from *E. faecalis* N48 strain, which was also *erm*B-positive, was included in the analysis. The details of the alignment are presented in Figure 3.

The aligned sequences of CA-MRSA ZY05 and CA-MRSA SO574/12 strains and a part of the pEflis48 plasmid, according to the progressive Mauve algorithm, were classified into one common LCB and recognized as homologues with a high level of similarity, coverage, and structure organization.

In the genome of strains ZY05 and SO574/12, the *erm*B gene was located in a chromosome on the mobile element structure (MES), as opposed to *E. faecalis* N48, where it was found on an independent extra chromosomal mobile genetic element (MGE), plasmid pEflis48. Chromosomally encoded *erm*B MESs demonstrated a high similarity to MES_PM1_ in CA-MRSA PM1 CC59/ST59/SCC*mec*Vb from Taiwan (since the genome sequence of *S. aureus* PM1 strain is deposited in GenBank database in the form of many parts, rather than a single molecule, it was impossible to include the PM1 genome in the whole-genome comparison analysis). Locally, small-scale multiple sequence alignment was performed, which is presented later in this publication. The *erm*B MES region in the SO574/12 clinical isolate was named an MES_PM1-like_ structure. It encodes 19 coding sequences and two insertion sequences. The appropriate MESs in ZY05 MRSA and pEflis48 MGE additionally contain disrupted *sat*-4 gene and gentamicin resistance predicted region but, according to the methodology used, do not encode the hypothetical protein that just follows down the ORF of *erm*B gene. A higher content of the mobile element protein gene was also noticed.

#### 2.2.4. Genetic Structure and Organization of *erm*B Carrying MES_PM1-like_ Structure in CA-MRSA SO574/12

According to the progressive Mauve algorithm, local multiple sequence alignment (MSA), nucleotide BLAST, PubMed, GenBank, and other NCBI databases analysis, the genetic structure and organization of the *erm*B-carrying region in the MES_PM1-like_ structure in the genome of CA-MRSA SO574/12 CC59/ST338/SCC*mec*Vb/PVL(+)/clone Taiwan clinical isolate was proposed. The visualization of the presented structure is shown in Figure 4.

The MES_PM1-like_ structure detected in contig 33 of SO574/12 was a 14,690 bp composite transposon consisting of 19 CDS and two 127 bp IS1216V-mediated direct repeats flanking the whole region. The *erm*B gene was encoded within the transposon Tn551 (5266 bp), at the 5′-end of the region as a part of *erm*B gene cluster, which consists of three ORFs: *erm*BL leader peptide (84 bp, 27 aa sequence); *erm*B (738 bp, 245 aa), and the 132 bp (43 aa-associated protein) ORF *erm*B-AP, located downstream the *erm*B gene. Tn551 also contains the mobile genetic element gene of transposase *tnp*551 (2919 bp, 972 aa) and invertase *tnp*R (555 bp, 184 aa) open read frame. The other three MES_PM1-like_ structures encoded antimicrobial resistance determinants, *aph*(3′)-III*—*aminoglycoside 3′-phosphotransferase; Δ*sat*-4*—*partly deleted streptothricin acetyltransferase and *ant*(6)-I*—*aminoglycoside 6-nucleotidyltransferase genes, are probably a part of the mobile element structure Tn5405. The other eleven genes of the MES_PM1-like_ structure are described in Figure 4.

The results from multiple local alignments comparing the nucleotide sequences of the four mobile genetic structures encoding the *erm*B gene cluster include the following: the MES_PM1-like_ structure of CA-MRSA SO574/12 SCC*mec*Vb/ST338/CC59 clinical isolate; MES_PM1-like_ structure of CA-MRSA ZY05 SCC*mec*Vb/ST338/CC59 reference Taiwan clone (GenBank accession number: CP045472.1); MES_PM1_ structure of CA-MRSA PM1 SCC*mec*Vb/ST59/CC59 reference clone (GenBank accession number: AB699882.1); and *Enterococcus faecalis* N48 plasmid pEflis48 (GenBank accession number: MT877066.1). In the case of MES_PM1_ and two MES_PM1-like_ structures in Taiwan clones from Taiwan, China, and Poland, the identity and coverage were equal to 100%, without any mismatches. The identification and coverage of homology sequence in plasmid pEflis48 were 99.99% and 99.15%, respectively, with one detected mismatch. The details of comparison are shown and described in Figure 5.

#### 2.2.5. Genomic-Based Antimicrobial Resistance Analysis

Fifty-four genes with well-predicted function related to the mechanisms of antibiotic resistance were detected. The genes are listed and characterized by the mechanism of resistance and the function of gene product in Table 5 and Appendix A. Determinants were divided into two essential groups: the first group containing genes directly correlated with the phenotype of resistance and the second group with genes that are either intrinsic or species-specific, or they encode the target of the drug in the tested genome, so their lack, derepression, or overexpression or other mutational changes demonstrate a resistance phenotype. The correlation of the detected genes with an antimicrobial resistance profile is presented in Table 2.

#### 2.2.6. Genomic-Based Virulence Factors Analysis

Sixty-five genes with product functional assignments were detected. Determinants were grouped into seven categories according to the function as a virulence factor: adherence factor (11 genes); antiphagocytosis (16); exoenzymes (9); immune evasion (5); iron/heme uptake (8); secretion system (5); and toxins/superantigens (11). Virulence factor genes in the MRSA SO574/12 strain were detected and analyzed (see Appendix A).

The strain was positively confirmed as a Panton–Valentine leukocidin, PVL gene carrier (*luk*S/F-PV), which was encoded on SA2_PM1-like_ phage. The nucleotide sequence of SA_PM1_ phage, originated from the *S. aureus* PM1 strain, showed 99.9% identity (data not shown).

## 3. Discussion

Here, for the first time in Poland, we present a detailed genomic characterization of a representative Polish variant of CA-MRSA ST338-SCC*mec*V_T_/CC59 PVL-positive clone, encoding the *erm*B gene cluster as a determinant of constitutive resistance to MLS-B antibiotics (cMLS-B). The main theme of this study, MRSA strain SO574/12, was isolated from an infected surgical wound in 2012. We have shown that it carries SCC*mec* type (subtype) Vb, also known as SCC*mec*V_T_ (5C2&5), cassette chromosome recombinase *ccr* genes complex type *ccr*C1-allele-2, *ccr*C1-allele-8, and *mec* gene complex class C2. The isolate represents the *spa*-type t437, *agr*-type I [29], sequence type ST338, and clonal complex CC59 and exhibits the *luk*S/F-PVpositive genotype. It is positive for the chemotaxis inhibitory protein *(chp* gene) but does not contain *sak* and *sep* virulence factors mediating immune avoidance functions. Based on all such extensive genomic analysis, the strain has been classified as an epidemic CA-MRSA clone “Taiwan”. ST338/CC59 is a single locus variant (within the *gmk* gene) derived from ST59/CC59 and has not been reported yet as a global pandemic clone. The first genome of ST338-SCC*mec*V/CC59 PVL-positive isolate has been published just recently in 2020 [5,30]. Representatives of this clone have so far been detected mainly in Taiwan, China, and several other Asia-Pacific countries, such as Japan, Vietnam, Singapore, and Australia [5,18,31,32]. Unfortunately, the patient’s medical history does not mention any travel in the period preceding the transplant procedure. In Europe, single cases have been reported so far in England, Denmark, The Netherlands, Norway, Sweden, Hungary, Germany, and also in Poland [4,26,33,34,35,36], which might be a more probable travel history. However, none of them were confirmed as the *erm*B-positive isolate.

Comparative analyses at the genomic DNA level, as well as analysis of the genetic structure and organization of the *erm*B-carrying region within the surrounding genetic environment, were carried out compared to the sequence of the most-detailed characterized strains CA-MRSA ZY05 and PM1, originating in China and Taiwan [17,30]. According to that analysis of whole genomic DNA, we have suggested that the Polish isolate of CA-MRSA ST338-SCC*mec*Vb/CC59 PVL-positive strain is closely related to the Taiwan clone, represented by the CA-MRSA ZY05 strain from China [30]. The SO574/12 strain presents 98.50% genome homology compared to ZY05. However, the Polish strain had been isolated four years earlier than ZY05, which strongly suggests the SO574/12 MRSA had originated from other parts of the world, not from the Asian regions.

The strain SO574/12 expresses a multidrug-resistance phenotype, highly similar to the previously described PM1 ST59-SCC*mec*V_T_/CC59 strain from Taiwan and the ZY05 ST338-SCC*mec*V_T_/CC59 strain from China [17,30]. Meanwhile, 13 ST338/CC59 strains among MRSA isolates obtained in China between 2014–2019, from human blood, were resistant only to erythromycin, clindamycin, and oxacillin [37]. The SO574/12 isolate is an older variant of ST338/CC59 CA-MRSA, but it appears more resistant than the more recent Chinese ones. The genetic features, which are transferred in MGE, are unstable and metabolic-cost consuming but play an important role as an adaptative factor. In the case of the lack of selective factor (e.g., antibiotic), the MGE-carried genes can be lost from bacterial cells [38,39]. This may depend on the local antibiotic policy. The global whole-genome alignment with the use of the progressive Mauve algorithm needs to be performed for the SO574/12 genome and the 13 ST338/CC59 MRSA genomes described by Jin et al., to assess their phylogenetic relation. Unfortunately, the genomic sequences of the Chinese isolates were not provided for an independent verification.

The antimicrobial resistance profile of SO574/12 correlates with the presence of the *mec*A gene and *bla*Z/*bla*R1/*bla*I gene cluster, which are responsible for resistance to β-lactams. Resistance to aminoglycosides is associated with the presence of aminoglycoside-modifying enzyme genes: *aph*(3′)-III and *ant*(6)-I. Resistance to tetracyclines is expressed due to MFS efflux pump genes: *tet*K and *tet*38 [20,40]. The cMLS-B phenotype correlates with the presence of *erm*B gene cluster consisting of the *erm*B gene, its leader peptide regulatory region *erm*BL and an additional *erm*B-AP CDS, which encodes a peptide of unknown function [41,42]. Until recently, the main determinants of *erm*A and *erm*C of the MLS-B phenotype have been dominant among Polish MRSA strains. The *erm*B gene has been most often found in Gram-negative rods, *Streptococcus* spp., and *Enterococcus* species [20]. This study is the first description of the *erm*B-positive MRSA strain from Poland.

We have shown that the CA-MRSA isolate SO574/12 has the *erm*B gene located on the chromosome within a mobile element structure, similar to the part of MES_PM1_ of the CA-MRSA PM1 ST59-SCC*mec*Vb/CC59 strain from Taiwan [17]. This genetic region has been detected in contig 33 of the SO574/12 genome and was named MES_PM1-like_ structure. It is a 14,690 bp composite transposon consisting of 19 CDSs and two 127 bp IS1216V-mediated direct repeats flanking both ends of the structure. Insertion sequence IS1216V, belonging to the IS6/IS26 family, is 809 bp in length with 18 inverted repeats. IS1216V is a typical enterococcal insertion sequence, rarely found in *S. aureus*, but up to five copies of IS1216V are located in MES_PM1_ and MES_6272-2_ of ST59 *S. aureus* [43]. Multiple local alignment comparison of nucleotide sequences of four mobile genetic structures encoding the *erm*B gene cluster, MES_PM1-like_ structure of CA-MRSA SO574/12 SCC*mec*Vb/ST338/CC59 clinical isolate, MES_PM1-like_ structure of CA-MRSA ZY05 SCC*mec*Vb/ST338/CC59 reference Taiwan clone, MES_PM1_ structure of CA-MRSA PM1 SCC*mec*Vb/ST59/CC59 reference clone, and *Enterococcus faecalis* N48 plasmid pEflis48, has shown that in the case of MES_PM1_ and two MES_PM1-like_ structures in the Taiwan clones from Taiwan, China, and Poland, the identity and coverage is equal to 100%, without any mismatches. In turn, the identity and coverage of the homologous sequence in the pEflis48 plasmid are 99.99% and 99.15%, respectively, with only one discrepancy detected. It is most likely that *erm*B-carrying MES_PM1-like_ structure had been transferred in a multi-stage evolutionary process between enterococci and *S. aureus*. Part of this process was by horizontal gene transfer from a strain similar to *Enterococcus faecalis* N48, via plasmid pEflis48-like. This plasmid is a mobile, self-replicating genetic element with a mosaic structure containing regions typical of both *E. faecalis* and *S. aureus* genomes. The other significant evolutionary stage was probably insertion sequence IS1216V-mediated bidirectional interspecies gene transfer via homologous recombination mechanism between plasmid-carried and chromosome-encoded gene clusters [43].

The *erm*B gene is encoded within the transposon Tn551 (5266 bp) [24], which is a part of the MES_PM1-like_ structure at the 5′-end of the region as a part of *erm*B gene cluster. This genetic region consists of three ORFs: *erm*BL leader peptide (84 bp, 27 aa sequence), *erm*B (738 bp, 245 aa), and 132 bp, the 43 aa-associated protein ORF *erm*B-AP located downstream from the *erm*B gene. Tn551 also contains a mobile genetic-element gene of transposase *tnp*551 (2,919 bp, 972 aa) and invertase *tnp*R (555 bp, 184 aa) open read frame. The other three ORFs of the MES_PM1-like_ structure encoded antimicrobial resistance determinants, *aph*(3′)-III—aminoglycoside 3′-phosphotransferase, Δ*sat*-4—partly deleted streptothricin acetyltransferase, and *ant*(6)-I—aminoglycoside 6-nucleotidyltransferase genes, are probably a part of mobile genetic element Tn5405 [44].

In our study, the *erm*B ORF is preceded by a 273 bp regulatory region that contains the regulatory 84 bp *erm*BL open reading frame, encoding the 27 amino acid leader peptides. The hypothesis of translation arrest on *erm*BL as a mechanism for *erm*B induction by erythromycin has been proven many times using an in vitro toe-printing assay. According to the literature, the *erm*B regulatory region contains one short leader peptide called *erm*BL with its ribosome binding site (RBS1), a non-translational loop-stem structure, and several *erm*B coding sequences, including its ribosome binding site (RBS2) [41,42,45,46,47]. The latest research carried out by Wang et al. suggests that a new leader peptide, *erm*BL2, exists in the *erm*B regulatory region [48]. Based on the premature termination mutation and alanine-scanning mutagenesis of *erm*BL2, researchers have shown that the N-terminus of *erm*BL2 is essential for the expression of *erm*B-dependent resistance to MSL-B drugs [48]. Therefore, the detailed function of *erm*BL2 requires further investigations.

A total of 54 genes for antibiotic resistance were detected in the SO574/12 genome. The mere presence of some of these genetic factors does not correlate with the resistance phenotype (see Appendix A). These are: *tca*A/*tca*B/*tca*R associated with resistance to teicoplanin; *cls*A, *gdp*D, *mpr*F, *lia*F/*liaR*/*lia*S associated with resistance to daptomycin; *rps*J (S10p) and *mep*A/*mep*R associated with resistance to tigecycline; *iso*-tRNA (*ile*S) and *fus*A genes associated with resistance to mupirocin and fusidic acid. Most of these genes are ubiquitous, internal, and species-specific, or regulator-dependent or drug-target genes. Their absence, derepression, or overexpression due to mutational changes indicate resistant phenotypes [2,23,49,50]. The lack of mutations correlates with susceptibility phenotypes (see Table 2).

The virulome of CA-MRSA isolates has been recognized as abundant, mainly due to presence of the genes, which are connected with their ability to produce a wide spectrum of exotoxins and superantigens (hemolysins, leukocidin PVL, and exfoliative toxin) [7]. Whereas, the virulome of HA-MRSA strains tends to contain more genes that determine their adhesive properties, immune avoidance factors, and exoenzymes [15]. In the analyzed genome of SO574/12 CA-MRSA, 65 virulence-associated genes have been detected. Of the seven functional groups, genes of toxins and superantigens are the most numerous and diverse. The *luk*S/F-PV genes, located on phage ΦSA2_PM1-like_, determine the ability to produce Panton–Valentine leukocidin. The presence of these genes is typical among community-associated MRSA isolates [12,13]. The CA-MRSA analyzed in this study emerged from the hospital environment, which unexpectedly and significantly changes the local epidemiological situation in hospital wards, where, until recently, only healthcare-acquired variants of MRSA were diagnosed. The adhesive properties of the SO574/12 strain are due to the presence of several genes encoding microbial surface components, recognizing the adhesive matrix molecules, MSCRAMM. The ability to synthesize the polysaccharide, poly-n-succinyl-β-1,6-glucosamine (PNSG), during infection is an important virulence factor based on the SO574/12 adhesive isolate. PNSG is critical for biofilm formation, allowing bacteria to adhere to one another and may also promote adherence to other molecules, such as extracellular matrix (ECM) components. This may act as an excellent environment for the formation of *ica*ABCDR-dependent biofilm, a natural biological membrane. Due to the lack of the *sdr*D gene, the virulome of SO574/12 isolate (2012) shows similarity to ZY05 CA-MRSA (2016) but differs from the strains analyzed by Jin et al. (2014–2019), which were described as negative in *clf*A, *clf*B, *eap*, *cna*, *sdr*C, *sdr*D, and *ica*D genes [37].

## 4. Materials and Methods

### 4.1. Bacterial Strain

Methicillin-resistant *Staphylococcus aureus* strain SO574/12 was isolated from a male adult patient hospitalized in a surgical unit of the Infant Jesus Clinical Hospital of the Medical University of Warsaw (Poland) for a routine diagnostic procedure. The strain was recovered in February 2012 from the pus draining from an infected postoperative wound. Clinical sample was inoculated on Columbia Agar plate supplemented with 5% sheep blood (BioMerieux, Marcy-I’Etoile, France) and MRSA Chrom Agar plate (BioMerieux). Incubation was performed for 24 h at 37 °C under aerobic conditions. The identification of the strain was performed with the use of automatic system VITEK2, BioMerieux (GP cassettes). After preliminary tests, the SO574/12 *S. aureus* isolate was archived and stored deeply frozen at −70 °C.

### 4.2. Phenotype Characteristics

#### 4.2.1. Confirmation of Resistance to Methicillin

Resistance to methicillin was confirmed using the disc-diffusion (DD) method with cefoxitin (FOX, 30 μg) (Oxoid, Basingstoke UK), according to EUCAST (European Committee on Antimicrobial Susceptibility Testing) recommendations [28,51].

#### 4.2.2. Detecting Resistance to Other Antibiotics

Resistance to a set of other agents was assessed with the use of the disc diffusion (DD) method and the E-test method. The former (by Oxoid) was applied for penicillin (P); amikacin (AK); gentamycin (CN); ciprofloxacin (CIP); levofloxacin (LEV); mupirocin (MUP); fusidic acid (FUS); tetracycline (TET), and the latter (by BioMerieux) for ceftaroline (CPT); vancomycin (VA); teicoplanin (TP); linezolid (LZD); daptomycin (DPC); tigecycline (TGC); and spectinomycin (SC), according to the EUCAST guidelines [28,51].

#### 4.2.3. Type of Regulation of Resistance to Macrolide, Lincosamide, and Streptogramin B (MLS-B) Antibiotics (Qualitative Method)

The inducibility of resistance to MLS-B antibiotics was performed with the use of disc diffusion (DD) D-shape zone method with erythromycin (E, 15 μg), clindamycin (DA, 2 μg), and lincomycin (MY, 15 μg) (Oxoid), according to the EUCAST guidelines [28,51].

#### 4.2.4. Resistance to MLS-B Antibiotics (Quantitative Method)

MIC (minimal inhibitory concentration) values were assigned, based on the E-test methodology (tested antibiotics were ranged between 0.016 to 256 mg/L), for erythromycin (E), azithromycin (AZ), clarithromycin (CH), and clindamycin (DA) (BioMerieux), according to the EUCAST recommendations [28,51].

### 4.3. Genotype and Genomic Characteristics

#### 4.3.1. Genomic DNA Extraction

Pure MRSA strain SO574/12 colonies were revived by culturing on nutrient agar. Bacterial genomic DNA was isolated with use of commercial Genomic Mini Kit (A&A Biotechnology, Gdansk Poland), in accordance with the protocol of the manufacturer. Quality and quantity of DNA were assessed in Eppendorf BioSpectrometer^®^ fluorescence (Eppendorf, Wesseling Germany), with the use of Quant-iT™ PicoGreen™ dsDNA Assay Kit (Invitrogen, CA, USA). DNA integrity was analyzed after 0.8% agarose gel electrophoresis. High-quality pure genomic DNA was stored at −20 °C until further analysis.

#### 4.3.2. Detection of Antibiotic-Resistance Genetic Determinants—Targeted PCR Amplification

The presence of *mec*A and *mec*C genes was determined with PCR and appropriate primer pairs, according to the procedure described previously [52]. The cMLS-B phenotype was verified with PCR. The *erm*A, *erm*B, *erm*C, *msr*A, *msr*B, and *lin*A/A’ resistance determinants were detected, according to the method described previously [53].

#### 4.3.3. SCC*mec* (staphylococcal chromosome cassettes mec) Assignment

The type/subtype of SCC*mec*, the type of *ccr* gene complex, and the class of *mec* gene complex were determined according to the procedure described by Okuma and Oliveira [54,55].

#### 4.3.4. Multilocus Sequence Typing (MLST)

Conventional MLST was performed based on an evaluation of seven housekeeping gene sequences (*arc*C, *aro*E, *glp*F, *gmk*, *pta*, *tpi*, and *yqi*L), according to the procedure described by Enright [56]. The sequence type (ST) and clonal complex (CC) were determined by small amplicon sequence analysis in a database available at https://pubmlst.org/bigsdb?db=pubmlst_saureus_seqdef, accessed on 1 June 2021. The evaluated isolate was classified as individual MRSA clone, based on the results of SCC*mec*, ST, and CC typing.

#### 4.3.5. Whole-Genome Library Preparation and Sequencing

Based on the previously isolated pure genomic DNA, tagmentation was performed using the NEBNext Ultra II FS DNA Library Prep Kit (Illumina, CA USA), in accordance with the protocol of the manufacturer. Accurate quantitation of the library was performed with the NEBNext Library Quant Kit for Illumina.

The draft genome was obtained through short-read bacterial whole-genome sequencing (WGS) on an Illumina MiSeq platform (Illumina Inc., USA). Paired-end 300 base-pair sequencing was done with the use of MiSeq reagent kit 2 × 300 cycles, targeting at least 100× genome coverage.

#### 4.3.6. Sequence Quality Verification, Trimming, and Assembling

Sequence quality metrics of the analyzed genome were assessed with FASTQC bioinformatic tool (http://www.bioinformatics.babraham.ac.uk/projects/fastqc/, accessed on 1 June 2021) [57]. Raw sequencing reads were trimmed for quality, and residual library adaptors were removed with the use of fastp software https://www.biorxiv.org/content/early/2018/04/09/274100, accessed on 1 June 2021 [58]. Cleaned Illumina reads were assembled using the following set of bioinformatic tools: Bandage v0.8.1; Pilon v1.23; QUAST v5.0.2; SamTools V1.3; Assembler SPAdes v3.12.0; and online platform PATRIC BBRC v3.6.9. (https://patricbrc.org/app/Assembly2, accessed on 1 June 2021) [59,60,61,62].

#### 4.3.7. Genome Annotation and Genomic Features Assignments

The genome strain of interest was annotated using the Genome Annotation Service (GAnS), which uses the RAST tool kit (RASTtk) to provide annotation of genomic features [63,64]. The GAnS uses the k-mer-based antibiotic-resistance genes (ARG) detection method, and assigns functional annotation to each ARG, broad mechanism of antibiotic resistance, drug class, and, in some cases, specific antibiotic it confers resistance to.

The type (subtype) of SCC*mec*, the type of *ccr* gene complex, and the class of *mec* gene complex were confirmed with SCC*mec*Finder (v1.2) service, available on an online platform (https://cge.cbs.dtu.dk/services/), accessed on 1 July 2021 [65,66]. Single locus typing of *spa*A gene was performed via *spa*Typer (v1.0) [67]; sequence type and clonal complex were assigned using MLST typing v2.0 [68] and database available at https://pubmlst.org/bigsdb?db=pubmlst_saureus_seqdef, accessed on 1 July 2021.

Other bioinformatic tools used included ResFinder (v4.1.0) [69,70] for antimicrobial resistance assignment; Virulence Finder (v2.0) [71] for virulence factor assignment; and Plasmid Finder (v2.1.0) [72], and ME Finder (v1.0.3) [73] for mobile genetic-element detection. These services were available on a platform of the Center for Genomic Epidemiology https://cge.cbs.dtu.dk/services/, accessed on 1 July 2021. Visualization of analyzed genome as a circular map was generated with the CGviewer server [74].

#### 4.3.8. Comparative Genomics and Mobile *erm*B-Carrying Genetic-Structure Analysis

A comparative analysis was performed and phylogenetic relationships with genomes and mobile genetic structures of other strains of the respective species of interest were assessed using the Genome Alignment Service, according to the progressive Mauve algorithm [75] and Phylogenetic Tree Building Service according to RAxML algorithm, available at a livestream platform, PATRIC BBRC (v3.6.9) [76].

Graphical display for the multiple alignments of nucleotide sequences was created and visualized using the nucleotide Basic Local Alignment Search Tool, nBLAST, and NCBI Multiple Sequence Alignment Viewer (MSA) v1.20.1, available at https://www.ncbi.nlm.nih.gov/tools/msaviewer/, accessed on 1 August 2021.

Other bioinformatic tools were also used for genomic data investigations: nucleotide local alignment nBLAST (https://blast.ncbi.nlm.nih.gov/Blast.cgi/), accessed on 1 August 2021, publication database PubMed https://pubmed.ncbi.nlm.nih.gov/, accessed on 1 August 2021 and also nucleotide, genes, or genomic international databases available at =the National Center for Biotechnology Information website https://www.ncbi.nlm.nih.gov/, accessed on 1 June 2021.

#### 4.3.9. Reference Genome, Plasmid, and Mobile Genetic-Structure Sequences

Comparative analyses were performed with the use of DNA sequences of the following strains: *S. aureus* ZY05 CC59/ST338/SCC*mec*Vb chromosome, complete genome, GenBank accession number: CP045472.1; *S. aureus* PM1 CC59/ST59/SCC*mec*Vb mobile element structure (MES_PM1_), GenBank accession number: AB699882.1; and *E. faecalis* N48 strain plasmid pEflis48 partial sequence, GenBank accession number: MT877066.1.

## 5. Conclusions

In this study, for the first time in Poland, we introduce a detailed genomic characterization of a representative Polish variant of the CA-MRSA ST338-SCC*mec*V_T_/CC59 PVL-positive clone, known as the Taiwan clone, encoding the *erm*B gene cluster as a determinant of constitutive resistance to MLS-B antibiotics. The analyzed SO574/12 strain was reported as an extremely rare and significant microbiological material, unique in Poland.

The analyzed CA-MRSA isolate emerged in a hospital setting, which has unexpectedly and significantly changed the local epidemiological situation in wards, where until recently only healthcare-acquired variants of MRSA were identified.

We demonstrated that the *erm*B gene, unique among *S. aureus,* was located on a chromosome within the MES_PM1-like_ structure, which also encoded aminoglycoside- and streptothricin-resistance genes. We also proved that the MES_PM1-like_ structure was a composite transposon, contained a smaller Tn551, and was flanked by direct repeats of IS1216V insertion sequences, probably originated from *Enterococcus* sp.

The *erm*B is preceded by the 273 bp regulatory region that contains the regulatory 84 bp *erm*BL ORF, encoding the 27 amino acid leader peptides. The latest research suggests that a new leader peptide, *erm*BL2, also exists in the *erm*B regulatory region. Therefore, the detailed function of *erm*BL2 requires further investigations.

## Figures and Tables

**Figure 1 ijms-23-08755-f001:**
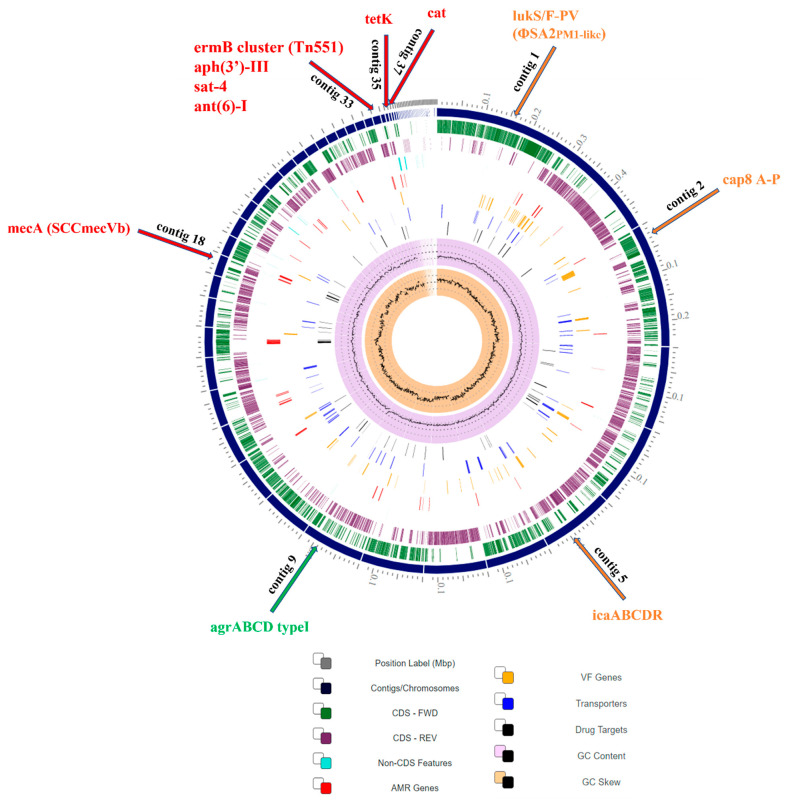
Circular genome view (CGV) of SO574/12 MRSA, with the essential genes’ assignment.

**Figure 2 ijms-23-08755-f002:**
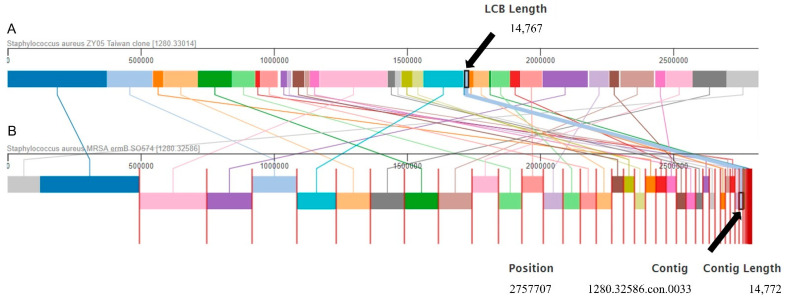
Visualization of global large-scale genome alignment according to progressive Mauve algorithm for genomes of CA-MRSA ZY05 and CA-MRSA SO574/12 strains. Each type of locally collinear block (LCB), is marked with a different color. The pair of LCBs contain the high conserved homology regions are assigned with the same stain. The homology regions that were defined within the aligned genomes are connected with the line in the same color. Both the contig 33 in genome of CA-MRSA SO574/12 isolate, which contains *erm*B gene, and its matching pair LCB in genome of CA-MRSA ZY05 strain are marked on the diagram in light blue.

**Figure 3 ijms-23-08755-f003:**
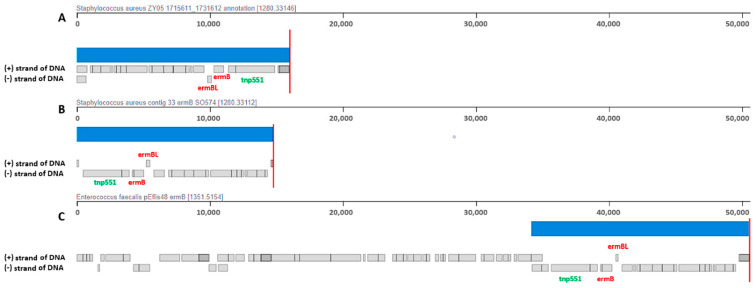
Small-scale local comparison of three *erm*B encoding structures and its genetic organization: (**A**)—*erm*B carrying LCB from CA-MRSA ZY05; (**B**)—*erm*B carrying contig 33 from CA-MRSA SO574/12; and (**C**)—*erm*B encoded region from *E. faecalis* plasmid pEflis48. Strips in the same blue color inform that aligned structures display each other as homologues and occur with a high level of similarity, coverage, and structure organization.

**Figure 4 ijms-23-08755-f004:**
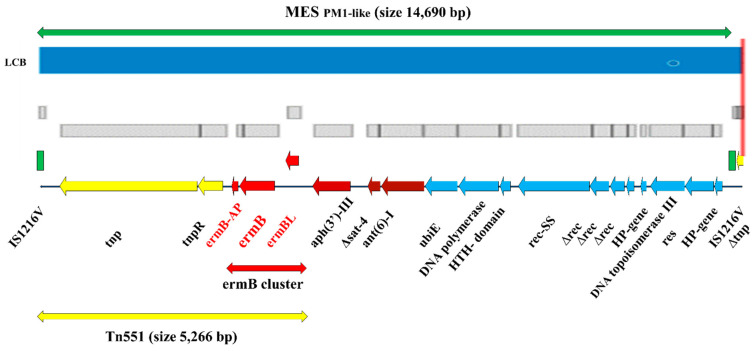
Genetic structure and organization of MES _PM1-like_ in genome of SO574/12 strain. MES—chromosomal mobile element structure; LCB—locally collinear block; IS1216V—insert on sequence IS1216V; *tnp*—transposase of Tn551 gene; *tnp*R—DNA-invertase gene; *erm*B-AP—*erm*B-associated protein (unknown function) gene; *erm*B—23S rRNA (adenine(2058)-N(6))-dimethyltransferase gene; *erm*BL—leader peptide region of *erm*B gene; *aph*(3′)-III—aminoglycoside 3′-phosphotransferase gene; Δ*sat*-4—partly deleted streptothricin acetyltransferase gene; *ant*(6)-I—aminoglycoside 6-nucleotidyltransferase gene; *ubi*E—methyltransferase gene, UbiE/COQ5 family; DNA polymerase—DNA polymerase gene, β-like region; HTH-domain—helix-turn-helix domain protein gene; *rec*-SS—site-specific recombinase gene; Δ*rec*—partly deleted recombinase gene; HP-gene—hypothetical protein gene; DNA topoisomerase III—DNA topoisomerase III; *res—*resolvase gene.

**Figure 5 ijms-23-08755-f005:**
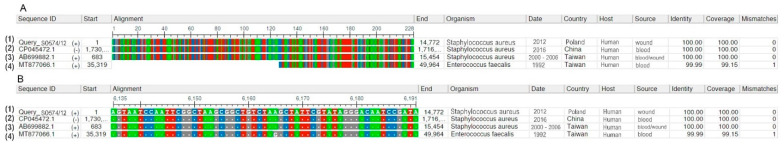
Multiple sequence alignment of *erm*B containing genomic regions (partial visualization of aligned sequences) of four strains: (**1**) MES_PM1-like_ of CA-MRSA SO574/12 SCC*mec*Vb/ST338/CC59 clinical isolate; (**2**) MES_PM1-like_ of CA-MRSA ZY05 SCC*mec*Vb/ST338/CC59 reference Taiwan clone (GenBank accession number: CP045472.1); (**3**) MES_PM1_ of CA-MRSA PM1 SCC*mec*Vb/ST59/CC59 reference clone (GenBank accession number: AB699882.1); and (**4**) *Enterococcus faecalis* N48 plasmid pEflis48 (GenBank accession number: MT877066.1). **Part** (**A**) shows alignment of 228 bp 5′-end of analyzed regions. In the case of strain number (1), (2), and (3), the nucleotide similarity and coverage were 100%; in the case of strain (4), the sequence was shorter (Δ127 bp), the identity 99.99%, and the coverage of the rest of DNA region was 99.15%, because of one point mutation (substitution). **Part** (**B**) shows alignment of internal regions of four analyzed strains to visualize transition mutation site 6165A:41,483G (CDS of *aph*(3′)-III gene).

**Table 1 ijms-23-08755-t001:** Essential phenotype and genomic features of MRSA SO574/12 clinical isolate.

Feature	Strain Characteristic
MRSA (gene)	Yes, low-level resistance (*mec*A)
MRSA phenotype	CA-MRSA
SCC*mec* type (subtype)	Composite Vb, also named V_T_ (5C2&5)
*ccr* genes complex type	*ccr*C1-allel-2, *ccr*C1-allel-8
*mec* gene complex class	C2
*spa* type (profile)	t437 (04-20-17-20-17-25-34)
MLST type (profile)	ST338 (19-23-15-48-19-20-15)
Clonal complex	CC59
International epidemic clone	Taiwan
MLS-B phenotype (gene)	cMLS-B (*erm*B)
AMR profile	E, AZ, CL, DA, MY, FOX, P, AK, TET
Main virulence factor (gene)	PVL (*luk*S/F-PV)
AGR TCSTS	*agr* ABCD type I
ACME	ND
Biofilm determination genes	*ica*ABCDR
Capsule determination genes	*cap*8 A-P
MGE (main carried gene):	
Plasmids	rep7a (*tet*K), rep7b (*cat*), rep19 (*bla*Z)
Tn	Tn551 (*erm*B)
IS	IS1216V, IS3-like family, IS200-like family
MES	MES_PM1-like_ (*erm*B, *aph*(3′)-III, Δ*sat*-4, *ant*(6)-I), SCC*mec* (*mec*A)
Prophages	ΦSA2_PM1-like_ (*luk*S/F-PV)

Legend: MRSA—methicillin-resistant *Staphylococcus aureus*; CA-MRSA—community-associated MRSA; SCC*mec*—staphylococcal chromosome cassette *mec*; *ccr*—cassette chromosome recombinase; MLS-B—macrolide, lincosamide, and streptogramin B group; cMLS-B—constitutive MLS-B; AMR—antimicrobial resistance; MLST—multilocus sequence type; ST—sequence type; CC—clonal complex; E—erythromycin; AZ—azithromycin; CL—clarithromycin; DA—clindamycin; MY—lincomycin; FOX—cefoxitin; P—penicillin; AK—amikacin; TET—tetracycline; PVL—Panton–Valentine Leukocidin; AGR—accessory gene regulator; TCSTS—two-component signal transduction system; ACME—arginine catabolic mobile element; MGE—mobile genetic elements; Tn—transposon; IS—insertion sequence; MES—chromosome-located mobile element structure; ND—not detected.

**Table 2 ijms-23-08755-t002:** (**A**,**B**) Antimicrobial resistance (AMR) profile of MRSA SO574/12 clinical isolate—phenotype with interpretation according to EUCAST guidelines [28] and correlation with AMR genomic-based data.

**(A)**													
Antimicrobial Resistance Profile—Strain MRSA SO574/12													
ATB results	Antibiotics, MIC (mg/L) or Diameter (mm)												
	Macrolide, lincosamide, streptogramin B group (MLS-B)						β-lactams			Aminoglycosides		Quinolones	
	E MIC >256	AZ MIC >256	CL MIC >256	DA MIC >256	MY 6	MLS-B test	FOX 16	P 17	CPT MIC 0.5	AK 17	CN 30	CIP 22	LEV 29
	HL-R	HL-R	HL-R	HL-R	HL-R	cMLS-B	LL-R	LL-R	S	LL-R	S	I	I
Gene	*erm*B						*mec*A	*bla*Z, *bla*R1, *bla*I	ND	*aph*(3′)-III	ND	*gyr*A, *gyr*B, *nor*A	*gyr*A, *gyr*B, *par*F
Phenotype expression	+ constitutive resistance						+	+	−	+	−	+	+
**(B)**													
ATB results	Antibiotics, MIC (mg/L) or Diameter (mm)												
	Glycopeptides				Oxazolidinone		Lipopeptide		Tetracyclines		Miscellaneous agent		
	VA MIC 1.5		TP MIC 0.75		LZD MIC 1.5		DPC MIC 0.25		TET 17	TGC MIC 0.094	MUP 35	FUS 30	SC MIC 48
	S		S		S		S		LL-R	S	S	S	S
Gene	ND		*tca*A, *tca*B, *tca*R		ND		*cls*A, *gdp*D, *mpr*F, *lia*F, *lia*R, *lia*S		*tet*K, *tet*38	*rps*J (S10p), *mep*A, *mep*R	*iso*-tRNA	*fus*A	ND
Phenotype expression	−		NEx		−		NEx		+	NEx	NEx	NEx	−

Legend: EUCAST—European Committee on Antimicrobial Susceptibility Testing; MIC—minimal inhibitory concentration, mg/L; HL-R—high-level resistance; LL-R—low-level resistance; I—intermediate; S—susceptible; ATB—antibiogram; ND—non detected; NEx—no expression; **Part** (**A**) E—erythromycin; AZ—azithromycin; CL—clarithromycin; DA—clindamycin; MY—lincomycin; MLS-B—macrolides, lincosamides, and streptogramin B group; cMLS-B—constitutive resistance to MLS-B antibiotics; FOX—cefoxitin; P—penicillin; CPT—ceftaroline; AK—amikacin; CN—gentamycin; CIP—ciprofloxacin; LEV—levofloxacin; **Part** (**B**) VA—vancomycin; TP—teicoplanin; LZD—linezolid; DPC—daptomycin; TET—tetracycline; TGC—tigecycline; MUP—mupirocin; FUS—fusidic acid; SC—spectinomycin.

**Table 3 ijms-23-08755-t003:** Main assembling and annotation data for genome of MRSA SO574/12 isolate.

Genome Feature	MRSA SO574/12 Characteristic
Size (bp)	2,792,694
G+C content (%)	32.74
Coarse consistency (%)	100
Fine consistency (%)	99.8
Completeness (%)	100
Contamination (%)	0
Number of raw reads	683,054
Total contigs count	67
Largest contig (bp)	494,917
Contigs N50 (bp)	127,953
Contigs N75 (bp)	65,061
Contigs L50	7
Contigs L75	14
Quast quality genome	good

**Table 4 ijms-23-08755-t004:** General genome and protein features of the MRSA SO574/12 and reference CA-MRSA ZY05 SCC*mec*Vb/ST338/CC59 (GenBank accession number: CP045472.1) strains.

Genome Feature	Clinical Isolate Genome of MRSA SO574/12	Reference Genome CA-MRSA ZY05 SCC*mec*Vb/ST338/CC59
Size (bp)	2,792,694	2,822,516
G+C content (%)	32.74	32.9
Protein encoding genes with functional assignment	1854	1844
Proteins encoding without functional assignment	789	778
tRNA genes	60	60
Protein coding sequences	2643	2622
Proteins with functional assignments	2131	2121
Hypothetical proteins	512	501
Total number of specialty genes:	330	314
Virulence factor genes (VFG)	98	95
Antimicrobial resistance genes (AMR)	80	72
Transporter genes (TG)	98	96
Drug target genes (DTG)	54	51

**Table 5 ijms-23-08755-t005:** Resistance genes and their regulatory regions, occurrence and expression correlate with the manifestation of the resistance phenotype.

Mechanism of Resistance	Gene	Gene Product/Function/KEGG Code	Class of Antibiotic	Antibiotics	PubMed ID
Antibiotic inactivation enzyme (transferases, hydrolases) and/or regulator modulating expression of antibiotic resistance genes	*ant*(6)-I	Aminoglycoside 6-nucleotidyl-transferase (EC 2.7.7.-), ANT(6)-I	Aminoglycoside	streptomycin	19603075; 2168151; 8293959
	*aph*(3′)-III/*aph*(3′)-IV/*aph*(3′)-VI/*aph*(3′)-VII	Aminoglycoside 3′-phosphotransferase (EC 2.7.1.95), APH(3′)-III/APH(3′)-IV/APH(3′)-VI/APH(3′)-VII	Aminoglycoside	butirosin, neomycin, kanamycin, amikacin, kanamycin, lividomycin, isepamicin, ribostamycin, paromomycin	6313476; 2846986; 2848443; 2550983
	*bla*I *bla*R1 *bla*Z	β-lactamase repressor BlaI β-lactamase regulatory sensor-transducer BlaR1 Class A β-lactamase (EC 3.5.2.6), BlaZ	β-lactams, Penicillins	penicillin, ampicillin, amoxicillin, piperacillin	9220009; 12591921; 6793593; 2555777
	*cat*A8	Chloramphenicol O-acetyltransferase (EC 2.3.1.28), CatA8 family	Phenicols	choramphenicol	15150221
	*sat-*4	Streptothricin acetyltransferase Sat-4	Streptothricins	streptothricin	31605529
Antibiotic resistance gene cluster, cassette, or operon, antibiotic target replacement protein	*mec*A	Penicillin-binding protein PBP2a, methicillin-resistance determinant MecA, transpeptidase	β-lactams	almost all β-lactams, except V generation of cephalosporins, e.g., amoxicillin, cefoxitin, ceftazidime, amoxicillin/clavulanic acid, meropenem	1507425; 1691614; 1544435; 30209034
	*fol*A/*dfr*C	Dihydrofolate reductase (EC 1.5.1.3)	Diaminopyrimidines	trimetoprim	8540692
Antibiotic target-modifying enzyme	*erm*B	23S rRNA (adenine(2058)-N(6))-dimethyl- transferase (EC 2.1.1.184), ErmB	Macrolides, Linosamides, Streptogramin B	erythromycin, azithromycin, clarithromycin, clindamycin, lincomycin, quinupristin, virginiamycin S, pristinamycin IA	11959553
Efflux pump conferring antibiotic resistance and/or the gene modulating antibiotic efflux	*bce*A *bce*B *bce*R *bce*S	Bacitracin export ATP-binding protein BceA Bacitracin export permease protein BceB Two-component response regulator BceR Two-component sensor histidine kinase BceS	Peptide antibiotics	bacitracin	25118291
	*sav*1866	Efflux ABC transporter, permease/ATP-binding protein YgaD	Ansamycins	rifampicin, rifaximin	18690712
	*tet*38	Tetracycline resistance, MFS efflux pump Tet38	Tetracyclines	tetracycline	26324534; 33619028
	*tet*K	Tetracycline resistance, MFS efflux pump TetK	Tetracyclines	tetracycline, doxycycline	7877638

Legend: KEGG—Kyoto Encyclopedia of Genes and Genomes.

## Data Availability

Not applicable.

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
