# Peer review of "Genomic Insights of First ermB-Positive ST338-SCCmecVT/CC59 Taiwan Clone of Community-Associated Methicillin-Resistant Staphylococcus aureus in Poland"

_ijms, 2022, doi:10.3390/ijms23158755_

Round 1
Reviewer 1 Report
Abstract
-Lines 12-17, I feel like there is a run-on sentence, could you please re-write this part to make it coherent.
-Line 18, beta-lactams should be changed to êžµ-lactams. Should be done throughout the manuscript.
-Lines 26 & 27, the authors need to clearly state what could this study add to what was already known on this topic. On other words, the significance of the study should be clearly delineated.
- Most importantly, you should also outline the key findings of genomic difference between the Polish MRSA and Asian counterpart.
Introduction
-In general, its too long and should be shorten especially the part of the differences between CA-MRSA and HA-MRSA.
-It would be more concise to move lines 33-52 after line 90. You also need to write 1-2 sentences about MRSA before jumping to line 54.
-Line 39, define [HA-MRSA].
-Line 50-53, references are required.
Results
-Table 2, you need to detail 2A and 2B in the legend separately. Citation also required.
Discussion
-In general, its lengthy and there is a reiteration for the results. The authors need to be careful.
Author Response
Dear Reviewer 1,
on behalf of the second author: Grażyna Młynarczyk and my own I would like to thank You for the reviewing and carefully evaluating of our manuscript. Referring to the suggestions and comments contained in the review, I would like to give You a response to, as follow below.
- Regarding the comments and suggestions on the “Abstract” section
“-Lines 12-17, I feel like there is a run-on sentence, could you please re-write this part to make it coherent.
-Line 18, beta-lactams should be changed to êžµ-lactams. Should be done throughout the manuscript.
-Lines 26 & 27, the authors need to clearly state what could this study add to what was already known on this topic. On other words, the significance of the study should be clearly delineated.
- Most importantly, you should also outline the key findings of genomic difference between the Polish MRSA and Asian counterpart.”
We re-wrote the abstract text to make it more coherent. We delineated the most important and key findings in our study.
We changed the “beta-lactams” phrase to “êžµ-lactams” throughout the text.
- Regarding the comments and suggestions on the "Introduction" section
“-In general, its too long and should be shorten especially the part of the differences between CA-MRSA and HA-MRSA.
-It would be more concise to move lines 33-52 after line 90. You also need to write 1-2 sentences about MRSA before jumping to line 54.
-Line 39, define [HA-MRSA].
-Line 50-53, references are required.”
We shortened and rearranged this section of the manuscript and made corrections as suggested. The references were also re-matched.
- Regarding the comments and suggestions on the "Results" section
“-Table 2, you need to detail 2A and 2B in the legend separately. Citation also required.”
We improved the Table 2 as suggested and added the citation.
- Regarding the comments and suggestions on the "Discussion" section
“-In general, its lengthy and there is a reiteration for the results. The authors need to be careful.”
We shortened this section of the manuscript and made corrections as suggested. We did our best to improve the quality and consistency of the text.
- We would like to add, we re-wrote also the “Conclusions” sections a bit to delineate the most important and key findings in our study and to make it more coherent.
The text of manuscript was also revised by translators, which ensured high quality and correctness of the English language.
All changes have been marked and highlighted for clarity and easy viewing. We submit our revised manuscript with tracked changes to facilitate the review in the file called “ijms-1820642_Revision_with_marked_corrections”.
I kindly request You to re-evaluate the revised manuscript.
Sincerely,
Ksenia Szymanek-Majchrzak
Reviewer 2 Report
In this work, the authors have merely isolated and characterized a previously known strain of Streptococcus (CA-MSR) from Poland that appeared to be a clonal variant of the Taiwan strain. Using genetic tools and bioinformatics the authors were able to identify antibiotic resistance genes and virulence factors that are necessary for pathogenesis of the pathogen including iron transport system and toxins. However, no new information comes up from this paper regarding the pathogen apart from the fact that its new geographic location. The paper is too elaborate and needs to be concise.
Author Response
Dear Reviewer 2,
on behalf of the second author: Grażyna Młynarczyk and my own I would like to thank You for the review and thorough evaluation of our manuscript. Referring to the suggestions and comments contained in the review, I would like to give You a response as follow below.
Referring to the suggestion, each section was shortened to make the manuscript more concise. All corrections can be viewed in “ijms-1820642_Revision_with_marked_corrections”.
The text of manuscript was also revised by translators, which ensures high quality and correctness of the English language
Dear Reviewer 2, for many years our research team has been providing the investigations of antimicrobial resistance profile, genetic background of the mechanisms of resistance and the epidemiology of MRSA strains isolated from patients hospitalized in one of the Warsaw Clinical Hospital. We had described and published the results of studies since early 1990s. The MRSA strain mentioned in our manuscript is the first one we have ever found and due to its uniqueness, we decided to carry out and describe its detailed characteristics in based on genomic DNA analysis. The Topical Collection "State-of-the-Art Molecular Microbiology in Poland" belongs to the section "Molecular Microbiology" of International Journal of Molecular Sciences is focused on a comprehensive overview of recent advances in molecular microbiology in Poland. In our opinion , the subject of our study is interesting and suitable enough to be published in intended Topical Collection of IJMS.
I kindly request You to re-evaluate the revised manuscript.
Sincerely,
Ksenia Szymanek-Majchrzak
Reviewer 3 Report
The authors have carried out the project to characterize both the phenotype and the genotype, based on whole genome sequencing for a detailed analysis of genomic features. The analyzed SO574/12 strain was reported as an extremely rare and significant microbiological material, unique in Poland, which was probably brought from the Asia-Pacific region.
The work is well designed and executed but the subject is not of general interest and the presentation is very long. My only suggestion is to make it a llitle short more and readable. For example tables 6 and 7 could be moved to supplementary results and additionally the conclusions to be more strong (1, 2 , 3 etc).
Another point is Legend to Figure 1 to be deleted once.
Author Response
Dear Reviewer 3,
on behalf of the second author: Grażyna Młynarczyk and my own I would like to thank You for the reviewing and carefully evaluating of our manuscript. Referring to the suggestions and comments contained in the review, I would like to give You a response to, as follow below.
Regarding to the suggestion, Table 6 and Table 7 were moved to the supplementary data and the Legend to Figure 1 was delated. Each section was shortened to make the manuscript more concise and readable.
The conclusions were re-written to make them stronger and more coherent. The most important and key findings in our study were delineated.
All changes have been marked and highlighted for clarity and easy viewing. I submit the revised manuscript with tracked changes to facilitate the review in the file called “ijms-1820642_Revision_with_marked_corrections”.
Dear Reviewer 3, I did my best to improve the quality and consistency of the article.
I kindly request You to re-evaluate the revised manuscript.
Sincerely,
Ksenia Szymanek-Majchrzak
Round 2
Reviewer 2 Report
The authors have significantly changed the manuscript. It now reads much better. They have cited all the work and it should be of interest to the readers.